# Physical Activity and Sleep Quality Association in Different Populations: A Meta-Analysis

**DOI:** 10.3390/ijerph20031864

**Published:** 2023-01-19

**Authors:** Haitao Zhao, Chuntian Lu, Cuixia Yi

**Affiliations:** 1Department of Sociology, School of Humanities and Social Sciences, Xi’an Jiaotong University, Xi’an 710049, China; 2College of Physical Education, ShanXi Normal University, Taiyuan 030031, China

**Keywords:** physical activity, sleep quality, different people, meta-analysis

## Abstract

The meta-analysis statistical methodology was used to test the effect of physical activity intervention on sleep quality. According to the preliminary results of the meta-analysis, moderating variables (age stage and physical activity intensity) were included for further study. On this basis, measures and schemes to improve sleep quality through physical activity were put forward. A preliminary Endnote X9 search of relevant literature from six electronic databases (Web of Science, Medline, PubMed, CNKI, PsycINFO and Wanfang) was performed. The results showed the following: (1) The overall test effect of physical activity intervention on sleep quality was not significant (*p* = 0.15). (2) Moderate- and low-intensity physical activity had significant effects on sleep quality (moderate intensity d = 2.56, *p* = 0.015; low-intensity d = 1.38, *p* = 0.03), while high-intensity physical activity had no obvious effect on sleep quality (d = 1.12, *p* = 0.32). (3) There were differences in the effect of physical activity on sleep quality at different ages. The effect of physical activity on sleep quality was more obvious in children and middle-aged and elderly people (children d = 1.24, *p* = 0.03; middle-aged and elderly people d = 1.98, *p* = 0.037), but not in young people (d = 1.32, *p* = 0.11). Conclusion: The overall effect of physical activity on sleep quality was not significant. Physical activity had an obvious effect on the sleep quality of children and middle-aged and elderly people but had no obvious effect on young people. Moderate-intensity physical activity had a certain effect on sleep quality, but high-intensity physical activity had no obvious effect on sleep quality. Future studies need to confirm these findings with a good large sample and moderating variables.

## 1. Introduction

Sleep quality, which is the most important indicator of sleep status, usually refers to problems such as difficulty in getting or maintaining sleep and the number of nighttime awakenings [1]. The most common problem with sleep quality is insomnia, which not only causes a certain degree of damage to individual alertness but also affects individual attention and memory, thus affecting the physical, psychological, social interaction and other aspects of insomnia [2]. Sleep quality directly affects people’s physical health, work efficiency and mental state [3]. Studies showed that people who sleep less than 6.8 h a day feel tired, weak and sleepy [4]. In the long run, this has adverse effects on the body and even leads to diseases, such as hypertension, heart disease, diabetes, endocrine disorders and a series of physiological diseases. At the same time, lack of sleep can also make people irritable, depressed, distracted, react slowly and display other mental illnesses [5]. According to the China Sleep Quality Survey (2022), the sleep index of Chinese people was 64.78 out of 100, indicating that their overall sleep condition was poor and often disturbed by sleep problems [6]. Therefore, the quality of sleep has become an important factor that affects human health and has been widely investigated by researchers all over the world.

Physical activity (PA) is defined as any form of body movement caused by muscle contractions [7]. Physical activity can improve the physiological, psychological and spiritual function of the human body, cause micro and macro changes in human organs, and achieve the purpose of the prevention and treatment of diseases [8]. Physical health improves when the energy expenditure from physical activity exceeds resting energy levels. Physical activity was shown to control depression and anxiety [9], increase serotonin levels in the brain [10] and boost immunity [11]. Proper physical activity not only helps to prevent common diseases, such as high blood pressure, heart disease and diabetes, but also regulates mood and reduces psychological symptoms, such as anxiety and depression [12]. Physical activity is also a simple, cost-effective non-drug therapy that improves sleep quality, and is considered a complementary alternative therapy for sleep disorders [13]. Because the body is physiologically and psychologically different in each age stage, the intensity of physical activity at different ages is the key to improving sleep quality.

The causes of sleep problems vary at different ages. Factors that interfere with the sleep quality of middle-aged and elderly people are due to degenerative changes in the body, reduced adaptability of the nervous system and reduced tolerance to changes in sleep time. Young people’s sleep quality is affected by mental pressure, life, social pressure, eating habits and other factors. The factors that affect children’s sleep quality are related to the development of the body’s internal clock and the influence of the external social environment [14]. Studies on improving sleep quality through physical activity have mixed results. Research suggested that people who are physically active have better sleep quality than those who are not [15]. Other studies found that moderate physical activity does not significantly improve sleep duration, habitual sleep efficiency or sleep disturbances [16]. In order to confirm the effect of physical activity intervention on sleep quality, as well as related studies on the moderating variables between interventions on sleep quality, this study adopted relatively mature meta-analysis research methods in the field of evidence-based medicine to evaluate intervention measures and test clinical validation effects.

Meta-analyses have been used to study the relationship between physical activity and sleep quality. Such as Rubio-Arias et al. explored the role of physical activity in improving insomnia in middle-aged women [17]. Lowe et al. systematically reviewed adults’ sleep disorders improved by physical activity [18]. Du et al. discussed the effect evaluation of traditional Tai chi physical exercise in the intervention of elderly sleep disorders [19]. All of these studies confirmed the role of physical activity in improving sleep quality, but few studies investigated correlations between the moderating variables and the effect of physical activity on sleep quality. Factors such as age category and physical activity intensity are important when formulating physical activity interventions for sleep quality. The lack of research content leads directly to the lack of planning physical activity programs and improving sleep quality.

In this study, meta-analysis theories and methods were used to conduct a meta-analysis of the published literature data and moderating variables related to the intervention of physical activity in sleep quality to provide a reliable basis for the study of the effect of physical activity on sleep quality.

## 2. Materials and Methods

This meta-analysis review was performed according to the Preferred Reporting Items for Systematic Review and Meta-Analysis (PRISMA) guidelines [20].

### 2.1. Information Sources and Search

In December 2021, we systematically searched six electronic databases (Sciencenet, Medline, PubMed, CNKI, PsycINFO and Wanfang) for literature on physical activity and sleep quality between 2016 and 2021. The literature search was performed by entering two groups of terms: (1) PA, physical activity, exercise, sport; (2) sleep quality, insomnia, sleep disorder. A total of 491 studies were obtained after the preliminary screening.

### 2.2. Eligibility Criteria 

According to the PICOS method of evidence-based medicine, the literature retrieval questions included in the criteria were divided into 5 aspects: participants, intervention, comparison, outcome and study [21].

The inclusion criteria of this study were mainly based on the PICOS method of circulating medicine, and the literature retrieval questions were divided into study object, intervention measures, control/comparison measures, outcome indicators and study design. According to the search questions, the literature inclusion criteria were formulated as follows: (1) the subjects were different groups of people with sleep quality problems; (2) a randomized controlled trial (RCT) was used; (3) we were able to use data (sample size, mean, standard deviation and *p*-value) to evaluate the literature on physical activity affecting sleep quality; (4) intervention measures: there were continuous (no less than 30 min of physical activity each time) with different forms of physical activity intervention (such as aerobic exercise, Tai chi, yoga, walking and other physical activities); and (5) the control group did not actively engage in physical activity or maintain their previous lifestyle. The exclusion criteria were (1) conference abstracts, papers and case studies; (2) subjects with serious medical conditions causing insomnia; and (3) the experimental group and the control group were used to conduct a two-component study.

### 2.3. Study Selection and Extraction

In terms of the paper screening, first of all, the retrieved literature was imported into the literature management software EndNoteX9 to delete irrelevant or repeated studies. Then, two investigators independently reviewed and screened the papers according to the full text, and the differences in the screening process were determined by seeking the views of a third party.

For the literature data extraction, two researchers independently extracted relevant indicators for the included literature. The extracted contents included the first author’s name, publication year, study location, sample size, sample category, average age, female proportion, study design, sleep evaluation index method, physical activity evaluation index method and correlation *p*-value. For incomplete or unclear information in the literature, we contacted the author through e-mail or other forms. If no reply was received after repeated contact, the paper was not included in the study. Figure 1 depicts the detailed process of study selection. A total of 491 records were initially identified and 11 randomized controlled trials were included for synthesized analysis [22,23,24,25,26,27,28,29,30,31,32].

### 2.4. Study Quality Assessment

The Physiotherapy Evidence Database Scale (PEDro) was used to evaluate the quality of the included studies. PEDro is composed of 11 questions; the first question is not included in the total score, while each of the rest of the questions provide a score of 1 or 0 points: meeting the corresponding criterion scores 1 point, not meeting the criterion scores 0 points. A study with a score ≥ 6 points can be considered to be of high quality. Two researchers rated the 11 articles according to the evaluation criteria. After the first round of evaluation, the items with score differences had their scores finally determined through group discussion (Table 1).

### 2.5. Quantitative Data Analysis

In this study, the bibliothemes were initially screened using Endnote X9, and 3599 subjects were tested using the inverse variance method of the fixed-effect model in Review Manager to analyze the indicators of the included bibliothemes. Since all indexes included in the literature were continuous variables with the same test unit, *p*-value and standard error (SE) were selected as the effect size indexes. When SE ≥ 0.8, this was considered a large effect size, 0.5 ≤ SE < 0.8 was considered a medium effect size and SE < 0.5 was considered a small effect size [33]. *I*^2^ was used to test the heterogeneity between different studies: when *I*^2^ = 0, it indicated that there was no heterogeneity between studies and the fixed-effects model could be directly used to combine the effect sizes; when *I*^2^ ≥ 50%, it indicated heterogeneity between studies. The random effects model was used to combine the effect sizes, and further subgroup analysis was conducted to find the source of the heterogeneity.

## 3. Results

### 3.1. Study Characteristics

In this study, 11 foreign studies on fixed effect trials were selected from 8 countries, namely, the United States, Germany, Spain, Ireland, Iran, Poland, Romania and Saudi Arabia. Participants included children; middle school students; college students; and young, middle-aged and elderly people. The sample size was 3599 participants, ranging in age from 4 to 58 years old, among which 9 studies included both males and females, 1 study included only females and 1 study included only males. The study designs were mainly cross-sectional. Sleep quality was evaluated using the PSQI (Pittsburgh Sleep Quality Index Questionnaire), AIS (Athens Insomnia Scale), CSHQ (subjective measurement of children’s sleep behavior and habits) and Wactigraph-bt (wristband detection sleep watch). The IPAQ (International Physical Activity Questionnaire), BSA-F (German Physical Activity, Exercise and Sports Questionnaire) and Wactigraph-bt (Wristwatch with Physical Activity Detection function) were used for physical activity evaluations (Table 2).

### 3.2. The Overall Effect of Physical Activity on Sleep Quality

The overall intervention effect size test for all samples of the 10 selected studies showed that physical activity had no significant effect on improving sleep quality (*p* = 0.15) (Table 3). The overall heterogeneity of the included literature was tested (*I*^2^ = 62%, *p* = 0.003). The fixed effect inverse variance model was used to combine the effect sizes. In the meta-analysis, there was a high degree of heterogeneity between groups of data, suggesting that other variable factors may have influenced the overall effect size (Figure 1). Therefore, the random effects model was required to combine the effects, and further subgroup analysis should be carried out to find the source of the heterogeneity.

The combined effect size of physical activity intervention on sleep quality was d = 0.02, and the 95% confidence interval was [−0.01, 0.05] (Figure 2). According to the standards set by Cohen for effect size, less than 0.2 is considered a small effect size. The effect size test showed that physical activity had no significant effect on improving sleep quality, and there were many influencing factors between the two. Overall, the intervention effect test showed a *p* > 0.05, indicating that physical activity had no significant effect on improving sleep quality in the collected data.

### 3.3. Regulating the Variable Subgroup Analysis

According to the heterogeneity of the global effect size test, subgroup analysis of the moderating variables was conducted to explore the source of the heterogeneity [34]. Subgroups of physical activity intensity and age were set for testing, and the test results were as follows. 

Intensity of physical activity: As some of the included literature did not classify the intensity of physical activity, only five articles of this moderating variable were included (two articles of low intensity, two articles of medium intensity and one article of high intensity). The 5 papers involved 3 groups of physical activity intensity (low-intensity, Moderate-intensity, high-intensity), with different intensities showing moderate heterogeneity regarding the effect size (*I*^2^ = 37.8%). Moderate-intensity physical activity had a larger effect size (d = 2.56, *p* = 0.015) regarding improving sleep quality, followed by low-intensity physical activity (d = 1.38, *p* = 0.03), and high-intensity physical activity had the lowest effect size (d = 1.12, *p* = 0.32).Age stage: According to the characteristics of different age stages in the included literature, the ages were divided into three stages: children (3–12 years), youth (13–35 years), and middle and old age (36–<60 years). The sample size of 1 study design was children and 8 studies involved young people (middle school students, college students and young people). Physical activity had a large effect size on the sleep quality of children and middle-aged and elderly people, and the effect was obvious (children: d = 1.24, *p* = 0.03; middle-aged and elderly people: d = 1.98, *p* = 0.037; young people: no significant effect (d = 1.32, *p* = 0.11)).

## 4. Discussion

### 4.1. Quality of the Included Literature and the Overall Effect Size

All the papers included in this study were carefully screened by the researchers. Studies that were duplicates, of low quality, missing data, conference reports and case studies were excluded from the 491 total papers, and 11 academic papers conforming to the inclusion criteria of this study were finally included in this study. Statistically, the overall effect of physical activity on sleep quality was not significant (d = 0.02, *p* = 0.15, 95% CI [−0.01, 0.05]).

### 4.2. Analysis of the Influence of Moderating Variables of Physical Activity Programs on Sleep Quality

Since the overall heterogeneity of physical activity intervention in sleep quality was moderately high (*I*^2^ = 62%), this study introduced moderating variables to further study the heterogeneity.

In terms of the intervention of physical activity intensity, physical activity intensity was taken as a variable, and three sub-variables, namely, low intensity, medium intensity and high intensity, were introduced for the heterogeneity and significance analysis. The results showed that the effect size of the three different intensity groups had moderate heterogeneity (*I*^2^ = 37.8%). Moderate-intensity physical activity had a larger effect size (d = 2.56) regarding improving sleep quality (*p* = 0.015), followed by low intensity (d = 1.38, *p* = 0.03), and high-intensity physical activity had the lowest effect size (d = 1.12, *p* = 0.32).In terms of age, according to the different age stages tested, the samples were divided into three age groups: children, young people, and middle-aged and elderly people. The results showed that physical activity had a larger effect on the sleep quality of children and middle-aged and elderly people. Among them, the effect was more obvious in the children group (d = 1.24, *p* = 0.03), followed by the middle-aged and elderly group (d = 1.98, *p* = 0.037), but the effect was not significant in the young people group (d = 1.32, *p* = 0.11).

### 4.3. An Analysis of the Effects of the Moderating Variables of Physical Activity Programs on Sleep Quality

In this study, the heterogeneity of physical activity intervention on sleep quality was high (*I*^2^ = 62%) and the effect size was small (d = 0.02). Therefore, moderating variables (age and activity intensity) were introduced to further explore and study the heterogeneity. The moderating variable is a variable that affects the relationship between dependent and independent variables. When the relationship between variable Y and variable X is a function of variable M, the relationship between Y and X is affected by the variable M. As moderating variables in this study, age type and physical activity intensity affected the direction and intervention effect of the relationship between physical activity and sleep quality.

Age category: Through statistical tests, it was found that children and middle-aged and elderly people were able to improve the quality of sleep via physical exercise, the effect of which was larger than for young people. Therefore, physical activity had a significant impact on the sleep quality of children and middle-aged and elderly people. However, among the included studies, there were two children and two middle-aged and elderly studies, each accounting for 20% of the total studies. Due to the small number of included studies, the research conclusions may be biased to some extent. Furthermore, the number of included studies and sample size of young people were large, but the effect of physical activity on sleep quality was not significant. This showed that physical activity affected sleep quality differently at different ages. Studies found that the association between physical activity and sleep quality changes with age. Children’s sleep quality problems are related to the developmental changes in the body’s internal clock and the influence of the external social environment. Different forms of physical activity, such as outdoor activities and sports games, can help to regulate the body clock and reduce sleep quality problems caused by social environments, such as noise and a bad mood [35]. Circadian rhythm changes are a major cause of insomnia in middle-aged and older adults. Middle-aged and elderly people who engage in appropriate physical activity can effectively improve their quality of sleep [36]. Young people’s sleep problems are due to study and work pressure, life and rest disorders, long-term mental work and other related factors. However, proper physical activity cannot effectively eliminate long-term mental stress, mental work, and irregular life and rest on the quality of sleep [37]. This result is basically consistent with the research conclusion of K. J. Reid et al. (2010) [38]. These studies also further confirmed the main reasons for the significant effect of physical activity on children and the middle-aged and elderly.Activity intensity type: It was found that moderate- and low-intensity physical activity had larger effect sizes, followed by high-intensity physical activity. The use of the three subgroups of moderating variables included in the literature was basically the same, and the number of samples with moderate intensity was the largest, followed by low intensity and high intensity. The analysis concluded that moderate- and low-intensity physical activity improved sleep quality better than high-intensity physical activity. Studies found that after moderate-to-low-intensity physical activity, certain amounts of sweat are eliminated. The sweating process can reduce orexin levels and concentration. Since orexin has the function of maintaining arousal and participates in the immune response, proper perspiration can reduce the arousal level by reducing the concentration of orexin, thus promoting sleep and improving sleep quality [39]. The research results showing that low- and medium-intensity exercise can effectively improve sleep quality were basically the same as the results of Rubio-Arias and other studies. It was confirmed that moderate- and low-intensity physical activity had a significant effect on improving sleep quality.

### 4.4. Research Limitations and Prospects

This study was a meta-analysis based on the existing literature, which may have been subject to some uncontrollable influences and limitations. The quality evaluation of alternative literature was only based on the subjective judgment of researchers. Although two rounds of paper screening were conducted, the quality of the papers was not evaluated using an objective quantitative method, and some studies may have had some biases. In the process of paper retrieval, screening and elimination, there will be some factors, such as missing selection and misjudgment, caused by subjective or retrieval technology reasons. For example, in terms of screening for sub-variable interventions, there are many viable sub-variables that were not included in the moderating variable content. Therefore, this research on moderating variables was restricted by the available sub-variable factors, thus affecting the comprehensiveness of the research results.

## 5. Conclusions

Through a literature search and meta-analysis, this study investigated the effects of physical activity on sleep quality. The results showed the following: (1) the overall effect of physical activity on sleep quality was not significant; (2) physical activity had a significant effect on the sleep quality of children and middle-aged and elderly people but had no significant effect on the sleep quality of young people; and (3) moderate- and low-intensity physical activity improved sleep quality significantly but high-intensity physical activity did not.

This study suggests that we should pay attention to the reasonable arrangement of physical activity, work and living environment in daily life, for example: (1) To improve children’s sleep quality using physical activities, mainly outdoor activities; sports games; and other low-intensity, relaxed and happy forms should be used. Furthermore, bright light and a noisy social environment should be avoided during sleep. (2) Physical activities that can be used to improve the sleep quality of middle-aged and elderly people are mainly low-intensity activities, such as outdoor walking, Tai ji chuan and Baduanjin. (3) Young people should appropriately adjust to the mental pressure of work, study and life; combine work and rest; and balance the daily physical and mental work to help improve the quality of their sleep.

## Figures and Tables

**Figure 1 ijerph-20-01864-f001:**
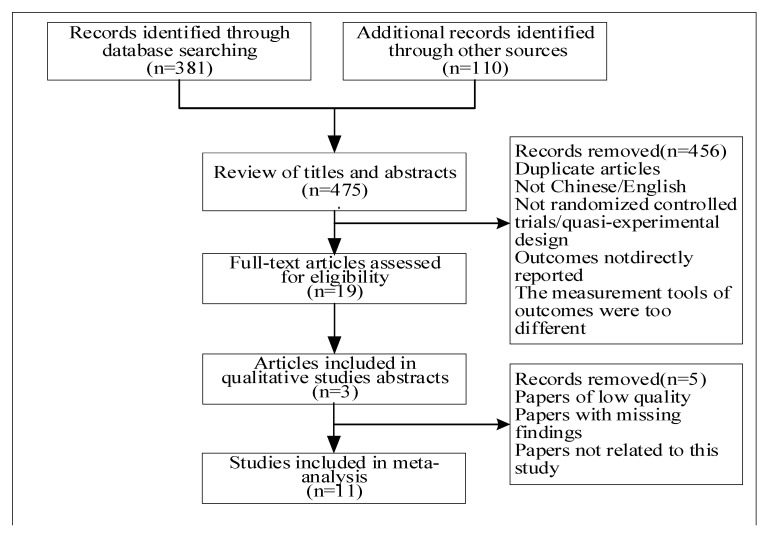
Flow chart of document selection and inclusion.

**Figure 2 ijerph-20-01864-f002:**
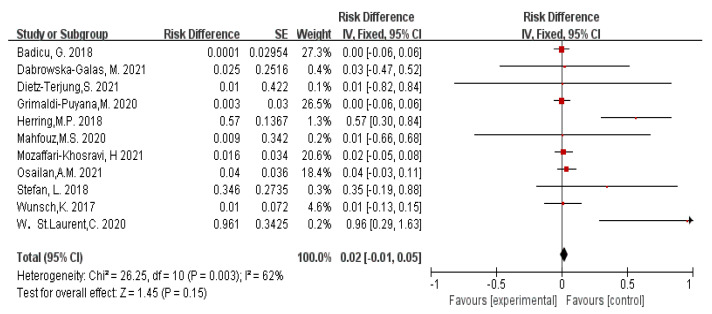
Forest graph showing the overall effect of exercise on sleep disorders [22,23,24,25,26,27,28,29,30,31,32].

**Table 1 ijerph-20-01864-t001:** PEDro score of the included studies.

Reference	Inclusion Criteria	Randomly Assigned	Allocation Concealment	Baseline Similarity	Subject Blinded	Blinding the Intervention Implementers	Blinding the Outcome Measure	At Least One Measurement	Intention-to-Treat Analysis	Statistical Report of En Groups	Results Point Measurements and Variation Measurements	Score
Badicu, G. (2018) [22]	1	1	1	1	0	0	0	1	1	1	1	7
Dabrowska-Galas, M. (2021) [23]	1	1	1	1	0	0	0	0	1	1	1	6
Dietz-Terjung, S. (2021) [24]	1	1	0	1	0	0	1	1	0	1	1	6
Grimaldi-Puyana, M.(2020) [25]	1	1	0	0	0	0	0	1	1	1	1	5
Herring, M.P. (2018) [26]	1	1	1	1	0	0	0	0	1	1	1	6
Mahfouz, M.S. (2020) [27]	1	1	0	1	0	0	1	1	1	1	1	7
Mozaffari-Khosravi, H (2021) [28]	1	1	1	1	0	0	0	1	0	1	1	6
Osailan A.M. (2021) [29]	1	1	1	1	0	0	0	1	1	1	1	7
Stefan L. (2018) [30]	1	1	1	1	0	0	0	0	0	1	1	6
Wunsch, K. (2017) [31]	1	1	0	1	0	0	0	0	1	1	1	5
W.St.Laurent, C. (2020) [32]	1	1	0	1	0	0	0	1	1	1	1	6

**Table 2 ijerph-20-01864-t002:** Characteristics of literature included in this study.

Reference	Study Location	Sample Size	Participants	Age (M ± SD)	% Female	Sleep Measure	*p*-Value
Badicu, G. et al. (2018) [22]	Romania	398	College student	20 ± 2.10	35%	PSQI	0.001
Dabrowska-Galas, M. (2021) [23]	Poland	80	Elderly	51.75 ± 5.57	100%	AIS	0.025
Dietz-Terjung, S. (2021) [24]	Germany	109	Young	31.8 ± 13.2	41%	Wactigraph-bt	0.010
Grimaldi-Puyana, M. (2020) [25]	Spain	306	Youth	20.7 ± 1.4	40%	PSQI	0.003
Herring, M.P. (2018) [26]	Ireland	418	Middle school student	15.1 ± 1.7	42%	PSQI	0.570
Mahfouz, M.S. (2020) [27]	Saudi Arabia	440	College student	22.38 ± 1.6	51.1%	PSQI	0.009
Mozaffari-Khosravi, H (2021) [28]	Iran	569	High school student	14.22 ± 0.88	47.8%	PSQI	0.016
Osailan A.M. (2021) [29]	Saudi Arabia	33	Youth	23 ± 1	0%	PSQI	0.040
Stefan L. (2018) [30]	Croatia	894	Elderly	80±3	56%	PSQI	0.879
Wunsch, K. (2017) [31]	Germany	64	College student	23.13 ± 5.12	67%	PSQI	0.010
W.St.Laurent, C. (2020) [32]	United States	288	Child	4.3 ± 0.8	47.2%	CSHQ	0.001

PR: Prevalence rate of sleep quality; AIS: Athens Insomnia Scale; CSHQ: Children’s Sleep Habit Questionnaire (a subjective measurement of children’s sleep behavior and habits completed by the guardian); Wactigraph-bt: a wristwatch monitor with a wrist detection function; PACE: Patient-Centered Assessment and Consultation, Exercise and Nutrition; BSA-F: German Physical Activity, Exercise and Exercise Questionnaire; IPAQ: International Physical Activity Questionnaire; *p*-value: statistical significance was defined at the level of <0.05.

**Table 3 ijerph-20-01864-t003:** Overall effect of PA intervention on sleep quality.

Number of Studies	Test of Heterogeneity			Effect Size and 95% Confidence Interval	Two−Tailed Test		
	X^2^	*p*	*I^2^*		Z	*p*	
11	26.25	0.003	62%	0.02 [−0.01, 0.05]	1.45	0.15	

## Data Availability

The raw data supporting the conclusions of this article will be made available by the authors, without undue reservation.

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
