# Peer review of "Physical Activity and Sleep Quality Association in Different Populations: A Meta-Analysis"

_ijerph, 2023, doi:10.3390/ijerph20031864_

Round 1

Reviewer 1 Report

Authors submitted a manuscript aiming to generalize the results of the study the associations between physical activity with sleep quality presented in previously publeshed works, using meta-analysis. Authors undertook research approach (meta-analysis) involving great responsibility. Generalization widely published works has usually great impact  on researches. While the topic is relevant, preparation of the manuscript has many flaws. Major revision, in my opinion, should improve the scientific soundness.

Abstract:

Clearly formulated problem and explicitly presented aim should be included instead of first sentence. Meta-analysis ought to be indicated as a statistical method. Moderating variables shoud be listed. Authors named the paragraph Conclusion, while presented results. There is lack of final conclusions. It should be suplemented.

Introduction:

Introduction is too laconic. Moderating variables (which one) mentioned in abstract are not included in introduction and their role in associations between physical activity and sleep quality is omitted. 

Metaanlysis ought to settle divergences between conflicting studies. Authors should clearly highlighted what divergences exist, and what they wanted to explain or summarize.

One of the main information is whether such a meta-analysis has been done before. Short indication should be included.

I don't know if authors used any meta-analysis guide, so I suggest to use any recognized guide, e.g. https://www.york.ac.uk/crd/SysRev/!SSL!/WebHelp/SysRev3.htm

Materials and methods

2.5 Quantitative Data Analysis

One of the key issues in meta-analysisis model selection: fixed effect or common effect. Although Authors specified which model they chose (fixed effect model), the assumption for this choice should be clearly presented. 

I suggest writing p (p-value) in lower case (line 97). 

An overview of the analyzed effects and types of figures (forest graphs) used in paper should be presented, what make it easier for the reader to follow the authors.

Results and conclusions

I suggest to use significant or not significant insted of obvious and certain (e.g. line 199, 202, 204; but also 15, 16 in abstract).

Conclusions should contain new hypothesis or (if not) what was limitation for statement. 

Reviewer 2 Report

The authors investigated physical activity (PA) and its association with sleep quality in different population samples. A meta-analysis was carried out using data extracted from relevant studies screened.  It was reported from the overall results that the effect of PA on sleep quality was not obvious.  However, it was shown from the analysis of co-variables that moderate and low-intensity PA affects sleep quality. Age was also reported to be a factor for sleep quality when PA was used as an intervention. Overall, it is an interesting work and I congratulate the authors for their effort in carrying out this research.  However, many issues need to be considered to improve the quality of the manuscript. I detailed my concern below.

Title

1.        Between - in. Appears to be redundant. Please revise

Abstract

2.       A brief overview of why the study was undertaken should be provided.

3.       The implication of the study findings should be captured at the end of the abstract

Introduction

4.       L27-30. Provide citations for such claims

Materials and Method

5.       L53-52. Mention the years of coverage

6.       L76. Was there any analysis to indicate the level of agreement on the choice variables between the two researchers? How exactly are the indicators decided?

Quantitative Data Analysis

7.       The description is too general, difficult to grasp how the data was retrieved and treated for analysis. A detailed description of the data retrieval, treatment and analysis should be provided.

8.       L76. Was there any analysis to indicate the level of agreement on the choice variables between the two researchers? How exactly the indicators decided

9.       Table 2: the p-values should be standardised. Some are reported in 3 decimals points while others are in 2 decimal places

Discussion

10.   L164-165. The authors indicated that the overall effect of PA on sleep quality was not obvious or significant. However, the p-value suggests otherwise (p =0.03).

11.   The discussion should be substantiated with the relevant literature. The authors merely summarised the results and no linkages or support was provided from other sources.

12.   A practical application section should be added to highlight the application and importance of the study findings to the community and the stakeholders involved

Reviewer 3 Report

Reviewer comments

Thank you for the opportunity to review this article.

This systematic review and meta-analysis investigated the relationship between physical activity and sleep quality across different age groups. Sub-group meta-analysis was also performed.

It is an interesting research topic, however, this article does not have a clear research question or study design, does not appear novel and lacks clear reporting.

The research question of this paper is unclear. The top of page 2 suggests the study investigates the impact of PA interventions on sleep quality, and whether PA improves sleep quality. However the title of the paper states “associations”, which would be a different approach. The research question requires more clarity.

The introduction does not explain the relationship between PA and sleep quality. Therefore, it is not outlined why this review is necessary. Furthermore, the authors state that the PA-sleep quality relationship has been explored previously, yet there is no disclosure of these findings and how the current paper adds to this literature. This further emphasises a lack of justification.

A closely matching review was published in 2021:        Wang, F. and Boros, S., 2021. The effect of physical activity on sleep quality: a systematic review. European Journal of Physiotherapy23(1), pp.11-18. https://doi.org/10.1080/21679169.2019.1623314

The reviewer suggests that the current article is too closely aligned with this previous paper for it to be published in IJERPH.

Other comments

Sleep quality is the chosen sleep outcome, however, this is not clearly defined in the introduction, and the reason for this choice is not disclosed. The introduction provides a review of general sleep outcomes, such as sleep duration and “lack of sleep”, however there are distinctions between sleep duration and sleep quality and their associations with health and well-being outcomes. In other words, the importance of sleep quality for the context of this paper is not made clear.

Since the research question is unclear, the study design and inclusion / exclusion criteria are more difficult to understand. Some of the inclusion / exclusion criteria also need re-considering. For, example, “2) there are ongoing, different forms of PA interventions” is very unclear. What study designs were the authors targeting? Experimental and observational or just experimental? Intervention studies only? Control groups? Is a pre-post design necessary (rather than cross-sectional)? This information should be reported and included in the search terms accordingly. In the flow chart (fig.1) it is reported that some studies have been excluded for not being RCTs / quasi experimental designs, however, the targeted study designs are not explained in the methods, so the reason for these exclusions are unknown. Furthermore, the study designs of the final sample are not disclosed in the results (other than “most were cross-sectional”) which are essential information for interpretation.    

The authors state “(3) Able to evaluate sleep quality and physical activity using subjective rating scales, showing sample size, mean, standard deviation and P-value in data.” However, studies with objective data have been included in the final sample and meta-analysis.

The discussion section essentially repeats the results section, with interpretation missing.

Overall, the reviewer recommends that the authors review the article by Wang, F. and Boros, S., 2021, and reconsider the necessity of the current article. An alternative approach could be considered, however, the authors must provide clear research questions, reasons for the papers relevance and necessity and a more robust study design and reporting approach.    

Round 2

Reviewer 1 Report

I accepted improvements and answers to my comments.

Reviewer 2 Report

The authors have adequately addressed my previous concerns.